# Can an Incentivized Command-and-Control Approach Improve Groundwater Management? An Analysis of Indian Punjab

**Sahil Bhatia *** and **S. P. Singh ***

Department of Humanities & Social Sciences, Indian Institute of Technology, Roorkee 247667, India
* Correspondence: sbhatia@hs.iitr.ac.in (S.B.); s.singh@hs.iitr.ac.in (S.P.S.)

**Abstract:** The Punjab Preservation of Subsoil Water Act 2009 is a legislative measure introduced to address the critical issue of groundwater depletion in Punjab, India. This research examines the implications of this Act and the rising groundwater scarcity in Punjab. Using qualitative research methods, including GIS mapping, it evaluates the postimplementation impact of the Act on groundwater conservation and water availability and assesses its effectiveness in achieving its objectives. This study reveals that the government's policies favoring wheat and rice have significantly contributed to the expansion of these crops, resulting in imbalanced agricultural practices. While the overall groundwater development in Punjab decreased from 170% in 2009 to 165% in 2017, a district-wise analysis reveals that the fall in the groundwater exploitation level in seven districts outperforms the rise in the exploitation level in the other thirteen districts of the state, showing overall minor or no improvement. This study proposes a multifaceted approach combining command-and-control measures with self-regulation incentives. It highlights the potential of incentivizing farmers to adopt sustainable practices, diversify crops, and implement water-efficient technologies. This paper also suggests the importance of involving stakeholders and the community in groundwater management, emphasizing the need for participatory approaches to ensure the long-term sustainability of water resources. While this study provides valuable insights, it is essential to acknowledge that its scope is limited to a qualitative assessment, and there may be challenges in generalizing the findings to all regions facing groundwater depletion.

**Keywords:** command-and-control measures; groundwater depletion; incentives; self-regulation

## 1. Introduction

Worldwide, the escalating problem of groundwater depletion caused by extensive agricultural withdrawals is generating mounting apprehension in specific regions, posing a significant threat to food security as presently, approximately 70% of global groundwater withdrawals, and an even larger proportion in arid and semiarid areas, are dedicated to sustaining agricultural practices [1]. Forty-four countries are ranked as being under high to extremely high baseline water stress, with most Middle East countries, Pakistan, and India classified as being under extremely high water stress, with the latter having both surface and groundwater stress [2]. The main reason behind this may be that the traditional common laws related to groundwater use and ownership that give landowners absolute rights over the resource are inadequate because they do not require individuals to protect or replenish this vital resource. As a result, some people use excessive amounts of groundwater for personal gain without considering the wider community's needs [3–6].

Despite the harmful consequences of this practice, no laws have been enacted to address it effectively [7]. In the case of India too, groundwater rights are still largely governed by the Indian Easement Act of 1882, the earliest evidence of groundwater laws [8]. This Act ties groundwater-usage rights to land ownership, giving users unlimited extraction rights [9]. The unregulated introduction of tube well technology led to India's first Groundwater Model Bill in 1970, with its revisions in 2011, 2016, and 2017, giving the individual

states the authority to pass laws and regulate groundwater usage [10,11]. While India's Groundwater Bill, amended in 2017, is a holistic way forward with the inclusion of decentralization, subsidiarity principles, and a bottom-to-top institutional framework, it has not significantly changed the situation of aquifers and groundwater exploitation. While most states were not bound to implement it, those that implemented it have performed so mainly in the case of drinking water and industrial usage [11,12]. Although groundwater is the primary source of drinking water, agriculture accounts for approximately 90% of water withdrawals in India [13,14]. The excessively exploited assessed units are primarily concentrated in India's northwestern region, encompassing areas such as Punjab, Haryana, Delhi, and Western Uttar Pradesh. Despite ample replenishable groundwater resources in the region, indiscriminate withdrawals have occurred, resulting in overexploitation [15].

The significant depletion of groundwater resources can be attributed to notable advancements in geological understanding, well-drilling techniques, pump technology, and the electrification of rural areas. The situation has been exacerbated by factors such as population growth, rapid urbanization, escalating per capita water-consumption rates, and the impacts of climate change [16–18]. This exploitation surged between 1950 and 1970, while parts of the developing world experienced a similar trend from 1970 to 1990 [19]. In the case of a developing nation like India, the introduction of the Green Revolution in the 1970s significantly increased food production by promoting groundwater-based irrigation. This approach led to the expansion of irrigated land and a notable surge in the adoption of electric and diesel pumps [20]. However, the widespread extraction of groundwater on a large scale has exacerbated detrimental consequences such as waterlogging, salinization, pollution, and a significant decline in water tables [21].

The initial comprehensive global assessment of groundwater depletion provided an estimate indicating that the total global groundwater depletion rose from approximately 126 ($\pm$32) km$^3$ per year in 1960 to around 283 ($\pm$40) km$^3$ per year in 2000 [22,23]. Another study reveals that approximately 80% (4.8 billion) of the global population in 2000 resided in regions of human water security threat [24]. Calculated through calibrated groundwater models, analytical methods, and volumetric budget analyses, global groundwater depletion (GWD) during 1900–2008 amounted to approximately 4500 km$^3$, equivalent to a sea-level rise of 12.6 mm, exceeding 6% of the total sea-level change (Konikow, 2011). From 2000 to 2010, the global GWD increased by 22%, reaching 292 km$^3$ in 2010, with significant increments in India (23%), China (102%), and the USA (31%). The majority of this GWD, approximately 83%, is associated with commodities, primarily in regions that are heavily reliant on overexploited aquifers, including the USA, Mexico, the Middle East, North Africa, India, Pakistan, and China, impacting major agricultural areas and population centers (Dalin et al., 2017). Certain aquifers, specifically the Ganges, Indus Basin, Californian Central Valley Aquifer System (USA), North China Aquifer System, and the Tarim Basin (China) show declining groundwater trends. These aquifers experience a situation where the withdrawal rates estimated through statistical analysis exceed those calculated by using Gravity Recovery and Climate Experiment (GRACE) estimates [25] (the data are generated by the GRACE satellite mission, which is a joint project by NASA and the German Aerospace Center (DLR)). More specifically, the water-depletion rate is estimated to be −19.2 ± 1.1 gigatons per year in Northern India; −5.5 ± 0.5 gigatons and −11.3 ± 1.3 gigatons per year in China's Xinjiang province and Beijing, respectively; −10.5 ± 1.5 gigatons per year in northwest Saudi Arabia; −32.1 ± 1.5 gigatons per year in regions including Turley, Syria, Iraq, and Iran; and −4.2 ± 0.4 gigatons per year in Southern California. These are all regions where there is agricultural activity and high groundwater demand [26]. The process of groundwater depletion can lead to several adverse consequences. These include diminished well yields; augmented pumping costs; the necessity to drill deeper wells; irreversible land subsidence; decreased base flow to springs, streams, and other surface water bodies; and the loss of wetlands [18,27]. Additionally, this will intensify competition within the water–energy–food (EWF) nexus. It also diminishes the groundwater quality through saltwater intrusion, further complicating

the situation [28]. Moreover, the unsustainable pumping of groundwater, leading to its depletion, can significantly impact a region's food and water security, ultimately adversely impacting marginal and small farmers, increasing socioeconomic inequality, and decreasing cropping intensity [20,29–31].

Australia and Spain serve as case studies outlining challenges in groundwater management due to their water supply challenges exacerbated by climate change, diverse irrigated agriculture, groundwater reliance, and decentralized catchment management. In the upper and lower Namoi groundwater source (Australia), overallocation resulted from government encouragement, excessive entitlements, and a lack of recharge data. In the Western Mancha aquifer (Spain), intensive, uncontrolled groundwater-based irrigation boosted economic growth but caused water table decline, impacting wetland ecosystems [32]. The declining US High Plains aquifer poses a threat to the region's irrigation-based economy. Each of the eight High Plains States adopts varying approaches to aquifer development and management due to differing state-water laws, creating challenges for integrated regional water management; however, this has also led to innovative solutions [33]. Water-related conflicts can emerge when water managers fail to provide essential supplies, as seen in Chennai, India, during the 2003–2004 drought, leading to private wells and water tanker systems [34]. Sudan possesses significant renewable and nonrenewable groundwater resources that are vital for various water needs. However, increasing demand, often unplanned, leads to issues like overexploitation and deteriorating quality. Effective policies and management are crucial for long-term sustainability. However, Sudan faces challenges, including a lack of data, poor understanding of aquifers, and fractured aquifers, complicating borehole placement and contamination risks. Shared nonrenewable aquifers, policy gaps, governance issues, capacity limitations, and coordination problems further hinder groundwater management [35]. Groundwater, often underestimated, plays a crucial role in supporting economic development, human well-being, and aquatic ecosystems in Africa. However, groundwater challenges, including pollution threats from various sources and the growing demand due to urbanization, industry, agriculture, and mining, are on the rise [36].

With the expansion of the human population and the depletion of natural resources, there is a growing demand to implement more stringent top-down management approaches for natural resources where individuals subject to regulation have limited motivation to enhance their performance voluntarily. The government can establish specific requirements and enforce compliance, making it an offense to fail to meet this obligation [37–39]. Command-and-control (CAC) regulations, which establish standards, oversee compliance, enforce adherence, and have been the primary approach employed by policymakers to regulate the environment, can take various forms, including environmental quality standards, permitted emission levels, and mandated or prohibited actions [37,40]. Such environmental regulatory programs heavily depend on legal frameworks and regulations. They are not without limitations, which become evident of excessive bureaucratic centralization, inflexibility, high costs, increased litigation, and delays [41]. However, in modern environmental management, particularly in the case of water-resources management, it is essential to consider four distinct spheres of action, which include command-and-control instruments, developing social consensus to define objectives and intervention plans, economic management instruments that encourage environmentally responsible behavior in a decentralized manner, and voluntary adherence mechanisms [42].

In water conservation, common CAC methods like rationing and technology standards tend to be inflexible and economically costly [43]. In California, CAC regulations have been the primary tool for governing water use, with residents paying a notably low price of USD 2 per cubic meter for potable water. This pricing lacks incentives to reduce consumption, and measures like limiting lawn watering to twice a week have been implemented [37]. The CAC policies are enforced through administrative regulations, including pumping limits for farmers, mining restrictions, and city-issued pumping permits. While criticized for their inflexibility and high costs, the CAC policies offer a diverse range of regulatory techniques that sometimes blur the line between CAC and incentive-based approaches [44,45]. A com-

parison was made between a mandatory low-flow appliance regulation and a moderate increase in water prices. The research utilized data from 13 California cities that rely on groundwater sources. The findings consistently indicated that, except for the least-plausible scenarios, raising water prices proved to be a more economically efficient approach than implementing technological standards for reducing the groundwater aquifer lift height over the long term [46]. To achieve their groundwater-management objective, policymakers in California have primarily relied on CAC regulations to govern water consumption in the state wherein the 'Water Conservation Plan of 2009' is an example of a CAC noneconomic regulation that mandates water suppliers and consumers to reduce water usage but offers limited or no mechanisms to facilitate the attainment of the set targets [37]. In the Israeli case of promoting green growth, the country addresses water scarcity and water-related environmental challenges through the implementation of advanced administrative and economic tools and incentives, resulting in significant political, structural, and economic transformations, such as the establishment of 56 Municipal Water Corporations; involving the private sector in public–private partnerships (PPP); and advocating for education, awareness, and water-saving campaigns to drive behavioral changes [47] (a cooperative arrangement between a government or public agency and a private sector company often used for the provision of public infrastructure, services, or projects). Critics of the CAC strategy advocate for a "smarter" approach utilizing market mechanisms, economic incentives, and voluntary compliance through environmental management systems (EMS), such as subsidies, permits traded in environmental marketplaces, and regular audits for regulatory compliance [48]. In the quest for regulatory alternatives to CAC, the promotion of self-regulation has been suggested as a means to enhance environmental performance, with the recognition that there exists a broader spectrum of policy options that lie between strict CAC and pure self-regulation [49]. To sustain water resources in heavily stressed aquifers, management practices need to be adjusted, often requiring reductions in pumping; however, top-down approaches face challenges in garnering support from affected irrigators, leading to growing momentum for community-based, bottom-up groundwater-management systems to ensure sustainability and avoid the imposition of top-down approaches [50–53].

In developing countries, bottom-up groundwater governance is frequently observed when a small number of smallholder farmers share a groundwater resource. In contrast, legal, institutional, and social barriers in developed countries often hinder collective groundwater governance among numerous large-scale industrialized farming operations [50,53–56]. Formal bottom-up groundwater governance can take two main forms: market-based approaches, employing financial incentives or penalties, and CAC approaches with restrictive pumping limits, while hybrid systems combining quotas with price limits or fees with pumping caps are also viable options [50,53]. For example, in Northwestern Kansas, C&C regulations with self-regulations have led to successful groundwater-management efforts, as irrigators collectively developed a conservation plan that includes reducing groundwater applications by 20% and implementing strict penalties, such as fines and water-rights suspension, for noncompliance with pumping limits [50,57,58]. Similarly, in response to excessive groundwater depletion affecting surface water supplies in Colorado's San Luis Valley, irrigators created six groundwater-management subdistricts based on hydrologic features. These subdistricts aimed to promote sustainable irrigation through economic-based incentives, such as a $45 per acre-foot groundwater pumping fee, to conserve water and compensate the surface-water-rights holders impacted by groundwater pumping. The revenue generated through these fees is used to subsidize the fallowing of irrigated cropland and restore the balance in the system [50,59]. California's Sustainable Groundwater Management Act (SGMA), implemented in 2014, mandates sustainable groundwater management through a command-and-control approach through incentives. It establishes Groundwater Sustainability Agencies (GSAs) in high- and medium-priority basins, responsible for developing and implementing sustainability plans. The SGMA also promotes the use of incentives, like grants and loans, to support local stakeholders in adopting sustainable practices. Empirical evidence indicates that the SGMA has enhanced

groundwater-resource management by facilitating improved monitoring, planning, and regulation efforts [60]. The Mexican policy of irrigation-management transfer in the 1990s involved the transfer of public control over irrigation districts to locally organized water-user associations (WUA) (community-based organizations or institutions formed by water users to collectively manage and govern water resources). This policy aimed to promote self-regulation and collective self-management among individual water users. However, the assumption that policy alone achieves its intended effects through governmentality can be challenged. It can be argued that while WUAs are presented as models for collective self-management, they are also subject to idealization through governmental technologies that shape their organizational identity [61]. In India, Haryana state has introduced a local CAC and incentive model, offering farmers an INR 7000/acre incentive to shift away from water-guzzling paddy cultivation. Additionally, permission to sow paddy is restricted in panchayat areas with a groundwater depth exceeding 35 m, particularly in 36 blocks where the groundwater-depletion rate has doubled over the past 12 years [62].

The transferability of successful groundwater-governance schemes to other stressed aquifer systems remains poorly understood, posing a challenge for developing conservation strategies, as the widespread application of similar rules may undermine long-term resilience by reducing institutional diversity, increasing the risk of misalignment with unique social and environmental contexts and contributing to recurring natural-resource-management issues [50,63–66]. This paper addresses the pivotal question of whether the Punjab Preservation of Subsoil Water Act 2009 has effectively mitigated groundwater depletion and encouraged responsible water usage. In doing so, it contributes to the existing groundwater-management literature by focusing on the postimplementation impact of the Act and its influence on agricultural practices in Punjab. Specifically, we aim to understand the consequences of the Act's measures on crop diversification, water conservation, and the overall sustainability of groundwater resources. This investigation also underscores the potential of combining command-and-control measures with incentives for self-regulation, offering a comprehensive approach that may be relevant to other regions facing similar challenges in groundwater management. In summary, this paper seeks to bridge the existing research gap by evaluating the efficacy of the Punjab Preservation of Subsoil Water Act 2009 and advocating for a holistic approach that includes participatory measures, financial incentives, and sustainable agricultural practices to promote responsible groundwater utilization and ensure its long-term sustainability.

## 2. Materials and Methods

This qualitative study applies a descriptive research design to investigate the impact of the Punjab Preservation of Subsoil Water Act 2009 on groundwater management in Punjab, India. It focuses on the state of Punjab, known for its significant groundwater exploitation and inefficient tube-well-irrigation practices. Approximately 75 percent of the 76.87 thousand hectares of gross cropped area in Punjab is irrigated through tube wells, with the rice–wheat cropping pattern accounting for 95 percent of the area dedicated to food crops and 86 percent of the total irrigated area [67]. As a result, Punjab is set to face significant food stress due to rising water scarcity [30,68,69], and the government policies of higher Minimum Support Prices (MSP) (the prices at which the government of a country agrees to purchase agricultural products from farmers) and power subsidies to farmers using electric tube wells are further complementing a monoculture in Punjab, with farmers mainly growing wheat in the winter and rice in the summer [70–72]. This study was conducted in four districts of Punjab—Sangrur, Jalandhar, Pathankot, and Bathinda—as they are predominantly groundwater-exploited (Sangrur and Jalandhar) and safe (Bathinda and Pathankot) categorized districts, which would help in comparing and analyzing the agricultural practices between the districts. The target population consisted of the farmers residing in these districts and irrigating their crops by using tube wells. The farmers were selected by using a multistage random sampling technique; in the first stage, the Sangrur, Jalandhar, Pathankot, and Bathinda districts were purposely selected. From each

district, 3 blocks were randomly selected, and from each block, 3–4 villages were randomly selected, and from each village, 8–10 farmers/households were targeted randomly to ensure the representativeness of the samples (Table 1). Observations and first-hand information were collected during the primary survey on farming practices, irrigation methods, and the implementation of the Punjab Preservation of Subsoil Water Act 2009. In addition, discussions were conducted with about 300 (246 for the final analysis) farmers to gain insights and feedback regarding their perceptions, experiences, and challenges related to groundwater management and regulations. The sample characteristics of the surveyed farmers are displayed below in Table 1.

**Table 1.** Sample characteristics of farmers surveyed in Punjab, India.

| District | Blocks | Villages | Number of Farmers Surveyed | Average Farm Size (Acres) | Average Rice Crop Yield (Qn/Acre) | Average Age of Farmers |
|---|---|---|---|---|---|---|
| Jalandhar | 3 | 6 | 58 | 21 | 3046 | 58 |
| Sangrur | 3 | 5 | 57 | 8 | 3067 | 51 |
| Bathinda | 3 | 9 | 76 | 13 | 2987 | 49 |
| Pathankot | 3 | 6 | 55 | 9 | 2576 | 55 |
| Total | 12 | 26 | 246 | 13 | 2919 | 53 |

By exploring these aspects, this paper aims to provide insights into the existing issues and the effectiveness and potential of evolving a mechanism combining command-and-control measures with incentivization and self-regulation in the context of Punjab's groundwater management. The secondary data for this study were collected from the Department of Economics and Statistics, Government of Punjab; the Central Groundwater Board, the Ministry of Agriculture and Farmer Welfare, the Government of India; and government-published reports. These sources provided valuable information on groundwater-management policies, historical data on groundwater levels, and agricultural practices in Punjab. The secondary data were complemented with key discussions with farmers.

The collected data underwent qualitative analysis by using themes derived from both secondary sources and field observations and discussions. The themes and categories were continuously reviewed and refined to ensure accuracy and consistency and to ensure the rigor and trustworthiness of this study. This study has several limitations. The sample size of the participants might not represent the entire population of farmers in Punjab, limiting the generalizability of the findings. Additionally, relying on secondary data sources introduces the possibility of data limitations and potential biases. However, it is important to acknowledge a potential Cultural and Language Bias that could have impacted the data-collection process. The primary survey was conducted in rural areas of Punjab, where farmers commonly use native Punjabi dialects that can vary significantly from urban Punjabi. This linguistic divergence can introduce challenges related to effective communication and data collection, potentially affecting the accuracy and comprehensiveness of the responses. To mitigate this issue, the assistance of local translators who were fluent in both the native rural Punjabi language and Hindi language commonly used in the survey instruments was employed. These translators played a vital role in bridging the linguistic and cultural gaps, ensuring that participants fully comprehended the survey questions and could respond clearly. While every effort was made to minimize the impact of a Cultural and Language Bias, it is essential to acknowledge that some degree of influence may persist, given the nuances of language and cultural practices. Nevertheless, it is believed that the involvement of local translators significantly contributed to overcoming these potential barriers and enhancing the accuracy and cultural sensitivity of our data-collection process.

## 3. Results

This section presents the main findings and outcomes of this study, which aims to shed light on various aspects of groundwater management in Indian Punjab. The results are organized into five main parts, starting with the present context of groundwater exploitation in Indian Punjab, where the current state of groundwater usage and its implications are analyzed. Next, a pre–post comparison of the 2009 Act in Indian Punjab to assess the impact of the Punjab Preservation of Subsoil Water Act 2009 on groundwater-management practices is conducted. Subsequently, this study delves into the Punjab Water Resources (Management and Regulation) Act 2020, examining its provisions and potential implications for groundwater governance. Following this, the issues while using the command-and-control approach as a policy instrument are addressed, and finally, an approach to regulating groundwater using the command–control and incentives is proposed, where a potential framework that combines command-and-control mechanisms with incentives to improve groundwater management in the region is presented.

### 3.1. The Context of Groundwater Exploitation in Indian Punjab

Despite having the highest rice yield and substantial production among major states, Indian Punjab is set to face significant food stress due to rising water scarcity [30,68,69]. Government policies of higher Minimum Support Prices (MSP) and power subsidies to farmers using electric tube wells are leading to a monoculture in Punjab, with farmers mainly growing wheat in the winter and rice in the summer [70–72]. Figure 1 shows the area under major crops in Punjab from 1980–1981 to 2020–2021. While the area under cotton, sugarcane, and maize has largely fluctuated but remained low, the area under wheat and rice has constantly increased. The area under wheat has increased from 2812 thousand hectares in 1980–1981 to 3530 thousand hectares in 2020–2021 (a net increase of 25.53%), while the area under rice crop has almost caught up to wheat from 1183 thousand hectares in 1980–1981 to 3149 thousand hectares in 2020–2021 (a net increase of 166.19%).

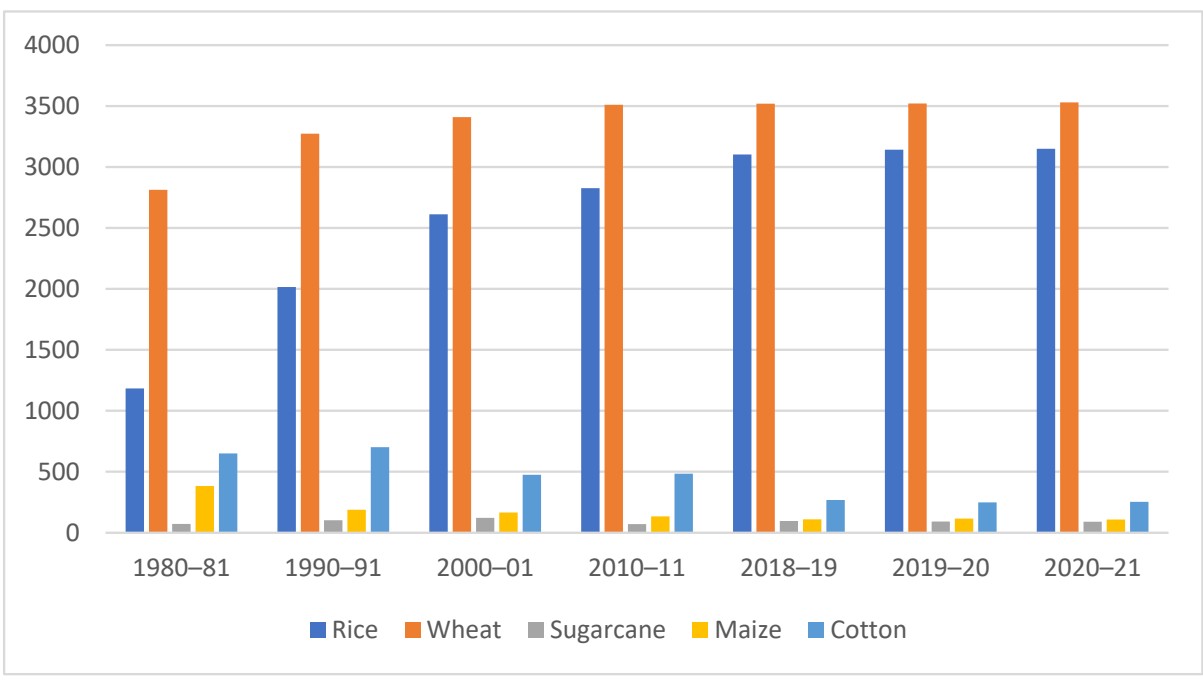

**Figure 1.** Area under major crops in Punjab, 1980–2020 ('000 hectares) [73].

This monoculture practiced by the farmers can be understood by looking at crop diversification, which refers to adding new crops on a particular farm and is intended to give a wider choice to expand production and minimize risk. Depending on the government policy, geoclimatic and socioeconomic conditions, and technological development in a

region, it is generally viewed as a shift from lower payoff crops to higher payoff ones [74]. The crop-diversification index ranges between 0 and 1. A higher crop-diversification index indicates a greater variety of crops grown within the system, implying a reduced dependence on a single or limited number of crops. Conversely, a lower crop-diversification index suggests more specialization in producing a few dominant crops. The Indian Economic Survey of 2018 [75] indicates that Punjab and Odisha share a common trend of declining crop diversification. The decline is sharpest for Odisha (−54%), followed by Punjab (−7.5%). The crop-diversification index of Punjab has witnessed a decline between 2005–2006 and 2014–2015. Heavy reliance on a limited number of crops (wheat–rice) has resulted in a decline in crop diversification over the years. During the mentioned period, there has been a decrease in the cultivation of alternative crops, such as pulses, oilseeds, fruits, and vegetables, contributing to a lower crop-diversification index. This declining trend raises concerns about the sustainability and resilience of agricultural systems in Punjab.

The Herfindahl–Hirschman Index (HHI), a widely employed metric for measuring market concentration, has been utilized to evaluate the level of crop diversification, assessing their differentiation in terms of crop selection [76–78]:

$$\text{Herfindahl} - \text{Hirschman Index (HHI)} = \sum NI = 1 - Pi^2$$

where Pi represents the acreage proportion of the ith crop in the total cropped area.

Table 2 presents the calculation of the crop-diversification index for Punjab using the HHI based on data from 2014. The index is determined by summing the squared market shares of each crop, where each crop's share is calculated as the square of its proportion in the total cropped area. The resulting HI value can range from 0 to 1, with higher values indicating a greater concentration or lack of diversification in crop production, while lower values indicate a more diversified crop portfolio or acreage. To interpret the data in line with the Economic Survey, the (1-HI) value was calculated. A higher (1-HI) value indicates a lower concentration or a more diversified crop production, suggesting a healthier agricultural system. Conversely, a lower (1-HI) value suggests a higher concentration or a lack of diversification, which may affect the overall resilience and sustainability of crop production in Punjab.

**Table 2.** Index of crop diversification, Punjab, calculated by using the HI index.

| Year/Crop Groups | Total Cereals | Total Pulses | Sugarcane | Condiments and Spices | Fruits and Vegetables | Oilseeds | Fibers | Total Cropped Area | 1-Herfindahl Index (HI) |
|---|---|---|---|---|---|---|---|---|---|
| 2014–2015 | 6539 | 16 | 97 | 1 | 183 | 48 | 422 | 7857 | 0.307 |
| 2015–2016 | 6618 | 20 | 92 | 0.9 | 190 | 49 | 336 | 7872 | 0.293 |
| 2016–2017 | 6670 | 20 | 89 | 1.3 | 182 | 45 | 285 | 7804 | 0.270 |
| 2017–2018 | 6702 | 14 | 97 | 1.6 | 164 | 36 | 292 | 7779 | 0.258 |
| 2018–2019 | 6743 | 14 | 95 | 1.2 | 195 | 37 | 268 | 7851 | 0.262 |
| 2019–2020 | 6795 | 10 | 91 | 0.8 | 187 | 37 | 248 | 7838 | 0.248 |
| 2020–2021 | 6801 | 10 | 89 | 0.6 | 187 | 37 | 252 | 7835 | 0.247 |

Source: author's calculation using land-use statistics, DES [67]; note: the area provided is in '000 hectares.

The declining crop diversification in Punjab presents a significant challenge that requires immediate attention. The monocropping pattern, primarily focused on paddy cultivation, has resulted in various environmental and economic consequences. The lack of crop diversification leads to an over-reliance on water resources and hampers the overall agricultural productivity and sustainability in the region [79–81]. To address this issue, the Punjab Preservation of Subsoil Water Act 2009 was introduced as a legislative measure to manage and conserve groundwater resources. Therefore, it is crucial to undertake a comprehensive pre–post comparison analysis of the Punjab Subsoil Water 2009 Act to

assess its effectiveness in addressing the declining crop diversification and groundwater depletion issues.

### 3.2. The Punjab Preservation of Subsoil Water Act 2009

One of the primary factors contributing to groundwater exploitation in Punjab is the practice of early rice transplantation (prior to mid-June), which leads to a significant depletion in groundwater. It is because the monsoon season is still some time away, the temperatures are very high, and the rate of evapotranspiration (ETR, the evapotranspiration rate is a crucial component of the Earth's hydrological cycle and refers to the rate of the combined processes of water evaporation from the Earth's surface and transpiration from plants or crops) is at its maximum [82,83].

In response to the increasing issue of rapid tube well expansion due to the onset of the Green Revolution, the Government of Punjab introduced the Punjab Preservation of Subsoil Water Act in 2009, which aimed at slowing down groundwater depletion. The Act came into force with an immediate effect when the legal notification was issued on 28 April 2009. It directly affects agriculture activities related to paddy. Before the Act was issued, farmers had no calendar restrictions regarding the plantation of paddy. However, since then, they have been directed to wait until 10 May and 10 June to sow and transplant their crops, respectively. Violations of the Act result in an inspection of the field by an authorized officer or a representative entering the area and instructing the farmer to destroy the crop if they are found guilty. The Act also describes a penalty of ten thousand rupees for a hectare of land in a month or a part of the penalty that could be imposed on the accused farmer [84]. The farmer can appeal to the collector within thirty days of the order passed by the inquiring officer describing the violations and the imposition of a fine. Any research project by the Punjab Agriculture University, Ludhiana, or any other research institutes identified through an official gazette by the state government shall be exempted from all such provisions of the Act. Since it was implemented to disallow farmers to plant the crop when evapotranspiration is at its maximum, any water-logged area in the state is also exempted from its provisions [84].

#### A Pre–Post Comparison of the 2009 Act in Indian Punjab

Studies suggest that the implementation of this Act has had a strong impact on reducing groundwater depletion. However, higher densities of tube wells per total cropped area and increased population density have contributed to a significant decline in groundwater levels [85–88]. Between 2009 and 2019, the average depletion of groundwater was approximately 8.91 m, with the most significant depletion of about 20.38 m in the Barnala district. The maps of the depth of the water table indicate that the proportion of the state's area with a water table depth greater than 10 m has increased from around 30% in 2000 to more than 75% in 2019 [87,88].

Thriving rice cultivation since the onset of the Green Revolution has led to the state facing sizable water stress recently. The fluctuation in groundwater levels reveals a concerning trend, as it has been observed to fall by more than 2 m during both the summer (May) and winter (November) seasons between 2012 and 2016 [89]. This decline in groundwater level highlights the severity of the water crisis in the region and raises alarms about the sustainability of current water-management practices. The same report in Punjab, Uttar Pradesh, and Assam also noticed a decrease in the annual water recharge. However, it has been attributed mainly to the change in the methodology, where the principle of the threshold value has been introduced to account for various factors, including rainfall recharge, changes in the area based on local revenue records, adjustments in parameters resulting from field studies, and other relevant considerations [89].

The dire groundwater situation in Punjab can be noticed through the Report of Dynamic Groundwater Resources Assessment of India by the CGWB in 2017 [89], which shows the state-wise groundwater development in the country. Punjab (165.77%), Rajasthan (139.88%), Haryana (136.91%), and Delhi (119.61%) are the states where the groundwater

resources have been overexploited (more than 100% groundwater development). It means that the groundwater extracted in these states is more than what can be annually extractable. Comparing this to the stage of groundwater exploitation for India, which is 63 percent and is still considered safe, few states lie in the critical and semicritical categories. In contrast, other states, including the northeastern states, lie in the safe category. Here, it is essential to distinguish between 'Physical' and 'Economic' scarcity of water. Physical water scarcity, prevalent in northwestern and southern parts of India, including Punjab, is when the demand for water resources exceeds its availability. On the other hand, the economic water scarcity observed in Central and Eastern India is due to water infrastructure that is too underdeveloped for its efficient use [90–93].

Table 3 shows the annual recharge and extractable groundwater resources in Punjab. The table displays that in a year, the monsoon and nonmonsoon groundwater recharge from rainfall and other sources adds up to the total annual recharge, which amounts to 23.93 billion cubic meters. Deducting the total natural discharges gives the annual extractable groundwater resource around 21.58 billion cubic meters. The other half of the table depicts that the present annual groundwater extraction in Punjab is estimated to be 35.78 billion cubic meters. Using this and the annual extractable groundwater resources gives us the stage of groundwater extraction, which is at 166 percent in Punjab, the highest among all Indian states. Comparing this with the 2009 groundwater resources shows that groundwater recharge has increased in Punjab over eight years, leading to an increase in extractable groundwater resources. It has led to a fall in the groundwater exploitation level in the state from 170 to 166 percent. The 2022 data further show the declining groundwater recharge and extractable groundwater. Since the current extraction of resources has also declined, the development level remains at 166 percent. However, it does not mean improving the water table level throughout the state.

**Table 3.** Groundwater resources of Punjab: annual recharge, extraction, extractable groundwater, and stage of groundwater extraction (in bcm).

| Assessment/Year | 2009 | 2017 | 2022 |
| --- | --- | --- | --- |
| Total annual groundwater recharge | 22.56 | 23.93 | 18.94 |
| Total natural discharges | 2.21 | 2.35 | - |
| Annual extractable groundwater resource | 20.35 | 21.58 | 17.07 |
| Current annual groundwater extraction | 34.66 | 35.78 | 28.02 |
| Stage of groundwater extraction (%) | 170 | 166 | 166 |

Source: Dynamic Groundwater Resources Assessment of India, 2022 [15,89], using GEC-2015 methodology.

Before the groundwater-resource comparison of the pre- and postimplementation of the Act is discussed further, it is essential to note the change in methodology used by the CGWB. Before 2017, all reports were based on the Groundwater Resource Estimation Committee (GEC)—1997, while the latest report is based on the GEC-2015 methodology [89]. The revised methodology involves a refinement in the norms for the variables, such as the specific yield, rainfall infiltration factor, canal, and irrigation recharge; use of a flag after an assessment for salinity, fluoride, and arsenic; and use of spring discharge data, if available, as a proxy for groundwater resources in hilly areas. The most significant change is regarding the categorization of the assessment units based on quantity. This change is summarized in Table 4.

The criteria for overexploited groundwater resources have remained constant, while for other categories, they have declined. Based on this, the district-wise groundwater development level comparison for 2009 and 2022 in Punjab is shown in Figure 2. In 2022 (Figure 2b), out of the 22 districts in Punjab, 18 were classified as overexploited as the level of groundwater development is more than a hundred percent. These are Gurdaspur, Amritsar, Tarn Taran, SAS Nagar, SBS Nagar, Fategarh Sahib, Patiala, Barnala, Sangrur, Kapurthala, Jalandhar, Moga, Ludhiana, Mansa, Hoshiarpur, Ferozpur, Bathinda, and

Faridkot. Nine of these eighteen districts, namely Sangrur (312%), Jalandhar (254%), Moga (235%), Kapurthala (226%), Barnala (220%), Patiala (216%), Ludhiana (216%), Patiala (216%), and Fategarh Sahib (207%) have a groundwater development level of more than two hundred percent, which shows the dismal situation of groundwater resources in these districts.

**Table 4.** Categorization of assessment units based on quantity: GEC 1997 and GEC 2015.

| Category of Groundwater Resource | GEC Methodology 1997 Used for Reports before 2017 | GEC Methodology 2015 Used for 2017 Report |
|---|---|---|
| Safe | ≤70% | ≤65% |
| Semicritical | >70% and ≤90% | >65% and ≤85% |
| Critical | >90% and ≤100% | >85% and ≤100% |
| Overexploited | >100% | >100% |

Source: report of the Groundwater Estimation Committee, CGWB 2015 [94]; note: the percentage represents the level of groundwater development in an assessment unit.

A closer district-wise comparison of the 2022 report [15] with the 2009 report [95] gives interesting results. While the overall groundwater development in the state declined from 170 percent in 2009 to 164 percent in 2022, it may be misguided if the district-wise analysis is not conducted. The groundwater development level of the Fazilka and Pathankot districts is not known in 2009 as they were formed in 2011. A comparison of both reports shows that out of the remaining 20 districts, 11 had higher groundwater exploitation levels in 2022 compared to 2009. The fall in the groundwater exploitation level in other districts outperforms the rise in the exploitation level of these eleven districts, showing overall no improvement. However, district-level data undermine the 2009 Act, which was built on the principle of improving the state's groundwater level or reducing its exploitation.

In 2022, Fazilka, Sri Muktsar Sahib, and Pathankot were the only districts classified as safe or semicritical based on the pre–post monsoonal decline. A significant decline in groundwater levels was observed in 85 percent of the state area from 1984 to 2016. The magnitude of this decline varies from region to region, indicating spatial variations in the extent of groundwater depletion [15,89].

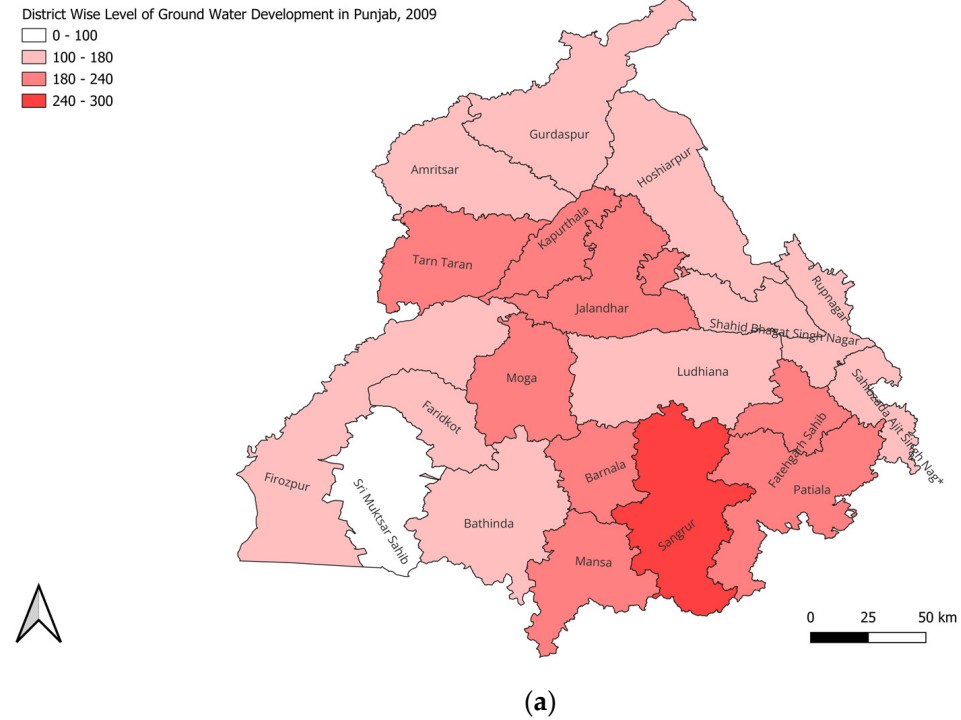

(**a**)

**Figure 2.** *Cont.*

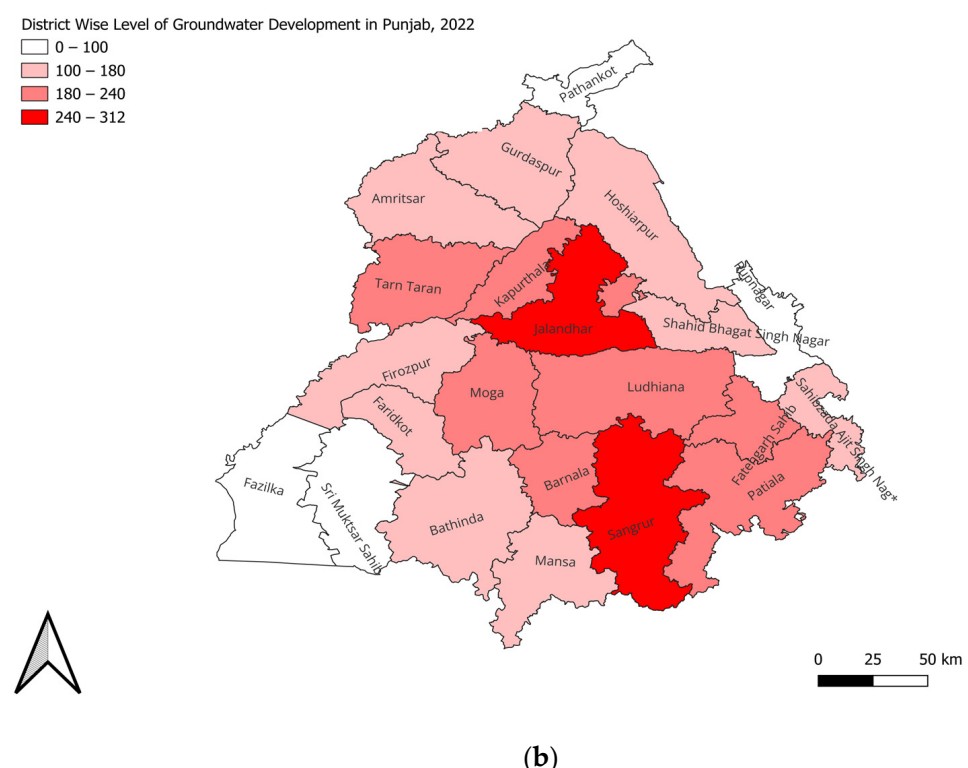

(**b**)

**Figure 2.** (**a**,**b**): district-wise level of groundwater development in Punjab; source: authors' own mapping based on the GGWB Report on Groundwater Resources of Punjab State, 2022; note: Pathankot and Fazilka districts were carved out of Gurdaspur and Ferozpur districts in 2011, respectively.

A district-wise pre- and post-Act analysis of the groundwater level depth was conducted by using GIS data for 2007 and 2017, considering both the pre- and postmonsoon levels and the maximum and minimum range of the water table. A high depth in the water table implies that a farmer needs to drill more beneath the surface to reach the water resource. Firstly, a postmonsoon comparison for the maximum water table depth range is made in Figure 3a,b. In 2007 (Figure 3a), districts like Kapurthala, Jalandhar, Sangrur, Patiala, Fategarh Sahib, and Rupnagar had the highest maximum water table depth of more than 26.7 m. In 2017 (Figure 3b), these districts maintained their dismal position, while other districts, such as Bathinda and Barnala, joined these districts as having the highest water table depth. Over ten years, districts such as Amritsar, Tarn Taran, Hoshiarpur, and Mansa also registered a higher water table depth. In contrast, districts such as Gurdaspur, Ferozpur, SBS Nagar, SAS Nagar, and Shri Muktsar Sahib maintained their position.

On the other hand, a premonsoon comparison shows that districts like Moga, Sangrur, and Barnala had the highest water table depth of more than 10.6 m in 2007. In 2017, these districts maintained their position, while Amritsar joined these districts as having the highest premonsoon water table depth. Over ten years, districts such as Rupnagar, Hoshiarpur, Bathinda, Faridkot, and Kapurthala also registered a significantly higher water table depth. In contrast, districts such as Gurdaspur, Ferozpur, Shri Muktsar Sahib, Mansa, Tarn Taran, Jalandhar, Ludhiana, and SBS Nagar maintained their position. Three districts, namely Patiala, SAS Nagar, and Fategarh Sahib, have the distinguished achievement of a significantly lower water table depth over ten years of study. A pre- and post-Act comparison shows Kapurthala, Sangrur, Rupnagar, Barnala, Amritsar, and Hoshiarpur to be the worst performers as they have the highest water table levels or registered significantly higher water table levels for 10 years for both pre- and postmonsoon levels. According to both reports, Jalandhar, Patiala, Fategarh Sahib, Tarn Taran, Mansa, Moga, Bathinda, and Faridkot also have high water table levels. While districts such as Gurdaspur, Ferozpur, Shri Muktsar Sahib, and SBS Nagar did not register any increase or fall in the water table,

Patiala, SAS Nagar, and Fategarh Sahib show an improvement in it. Data for Pathankot and Fazilka remain unavailable for comparison due to their formation in 2011.

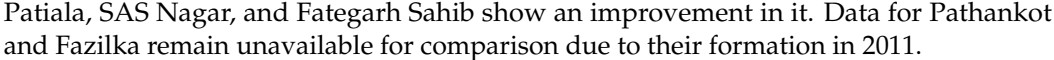

(**a**)

(**b**)

**Figure 3.** (**a**,**b**): district-wise maximum postmonsoon groundwater level in Punjab 2007 and 2017 (meters); source: author's own mapping using Statistical Abstract of Punjab, 2020 [96]; note: Pathankot and Fazilka districts were combined with the original districts they were formed from, as the data for the new districts were not given; data for the maximum water table depth of the Moga district are not available.

Table 5 shows that the number of overexploited, dark, and white blocks remained almost stagnant, barring one block in each category. The number of white blocks resembling safe blocks has fallen over the years. These three blocks are now in the grey category resembling semicritical resources. However, this comparison paints a partial picture of an otherwise miserable situation in the state as the number of overexploited blocks has doubled since 1984. Despite the measures taken, the water table has not yet improved. Similarly, the number of safe blocks has also fallen over the period.

**Table 5.** Overexploited and dark blocks in Punjab, 2009 and 2017.

| Blocks/Year | 2009 | 2017 |
|---|---|---|
| Overexploited | 110 | 109 |
| Dark | 3 | 2 |
| Grey | 2 | 5 |
| White | 23 | 22 |

Source: the GGWB Report on Groundwater Resources of Punjab State, 2022 [89].

The percentage of blocks overexploited in each district of Punjab according to the CGWB is shown in Figure 4. Twelve districts out of twenty were classified as having 80 to 100 percent of their blocks overexploited. Bathinda and Hoshiarpur have 43% and 40% of their blocks overexploited, respectively. The districts where the groundwater level has been least exploited in most blocks include Pathankot (0%), Shri Muktsar Sahib (0%), and Fazilka (25%).

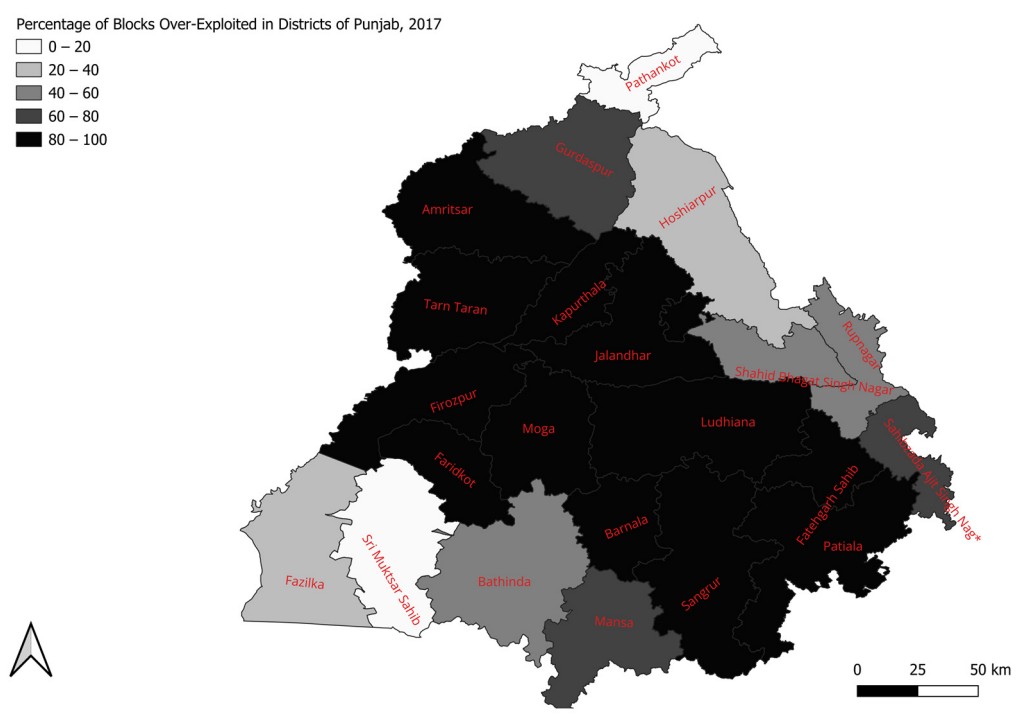

**Figure 4.** District-wise percentage of blocks overexploited, 2017; source: authors' own mapping using CGWB Report on Groundwater Resources of Punjab, 2017.

The support that the farmers have received to produce rice by using groundwater resources since the Green Revolution can be attributed to low investments in large irrigation projects and cheap electric pumps. Additionally, no or low charges for farm electricity operations have added to the groundwater woes in Punjab [68,97–99]. Figure 5a,b shows the district-wise areas under rice crop and tube well irrigation, respectively. The area under rice production in 2018 was mainly concentrated towards the middle portion of the state in districts like Ludhiana, Sangrur, Patiala, Ferozpur, and Tarn Taran. The significant

area under the crop also exists in Gurdaspur, Amritsar, Jalandhar, Moga, Bathinda, and Shri Muktsar Sahib. Similarly, a huge portion under tube well irrigation is observed in these districts. Sixteen districts have registered an increase in the area under rice crops between 2009 and 2018. The largest increase in the area comes from Sri Muktsar Sahib (82.11%), Bathinda (64.95%), Mansa (50.70%), Hoshiarpur (25%), and Faridkot (21.05%). An intriguing observation is that in 2009, all these districts had less than 100 thousand hectares of land dedicated to rice cultivation, indicating a significant untapped potential for expansion in both crop area and pressure on groundwater resources. The state's area under rice increased by 12.07% from 2735 thousand hectares in 2009 to 3142 thousand hectares in 2019 [67]. This means further pressure on the state's groundwater resources.

The Punjab Preservation of Subsoil Water Act 2009 focuses on only a single aspect of improving farm utilization of water–crop scheduling to reduce evapotranspiration. With emerging markets for water-extraction technologies since the Green Revolution, managing the demand for groundwater resources has become significant. No such demand-management measure of water resources has been mentioned in the 2009 Act. In the case of paddy cultivation, technologies like laser land leveling and conservation tillage methods can prove useful in reducing aquifer depletion [100–102]. If the latter is promoted, it can also reduce stubble-burning instances, which have troubled neighboring states like Haryana and Delhi. The government has allocated INR 1000 crores to promote crop diversification, focusing on increasing the cultivation of aromatic paddy basmati, cotton, and oilseeds while reducing water-intensive paddy cultivation. This allocation includes a revolving fund for basmati procurement, a 33% subsidy on cotton seeds, and incentives of INR 1500 per acre for farmers adopting the direct seeded-rice method. Additionally, the government plans to unveil an agriculture policy to conserve natural resources and support tree planting through financial assistance under the National Horticulture Mission Scheme [103]. It will involve a change in the policy of Minimum Support Prices and encouraging farmers to conserve groundwater through valuation. While the central districts of the state face rampant groundwater depletion, the southern and southwestern districts face extremely poor groundwater quality. Since both groundwater scarcity and poor quality adversely affect farm productivity, it is crucial to address quality issues if food security is to be secured. Pathankot is the only district in the state which, as of now, is the best performer regarding both measures of quantity and quality [104,105].

Studies since 2009 have shown varying results regarding the impact of the Act on groundwater levels. While some studies state that the Act can help reduce groundwater table exploitation by 30cm as the early transplanting of rice accelerates groundwater depletion, others find that a higher share of tube wells and increasing population density have led to declining water tables in the state and leaves the conclusion depending on the monsoon rain. Since rainfall in the state is variably distributed, and rice crops demand more water application than other crops, the importance of groundwater irrigation cannot be overestimated [83,85–88,106–108]. Hence, addressing the externalities arising from groundwater exploitation is extremely important. These include mass exploitation, increasing extraction costs for marginal and small farmers, and the absence of well-defined property rights, among others [109]. Regarding cropping patterns, Tripathi (2016) reports an increase in the total cropped area under paddy even after introducing the Act in 2009 [88].

As Punjab is ranked third in rice and wheat production and third in overall food grain production [110], a policy specifically for the marginal and small farmers must be framed. Because of increasing fragmentation, farm sizes have become more unequally distributed. It is difficult for governance and policy measures to reach those marginal land holdings. Small and marginal farmers face the maximum brunt of groundwater exploitation and its externalities [30,111].

The Punjab Water Resources (Management and Regulation) Act 2020, was enacted to complement and strengthen the existing legislation, the Punjab Preservation of Subsoil Water Act 2009, addressing certain gaps and evolving challenges in water-resource management. While the Punjab Preservation of Subsoil Water Act 2009 primarily focuses

on regulating groundwater extraction through paddy-crop scheduling, the Punjab Water Resources (Management and Regulation) Act 2020 aims to provide a more comprehensive framework for integrated water-resources management.

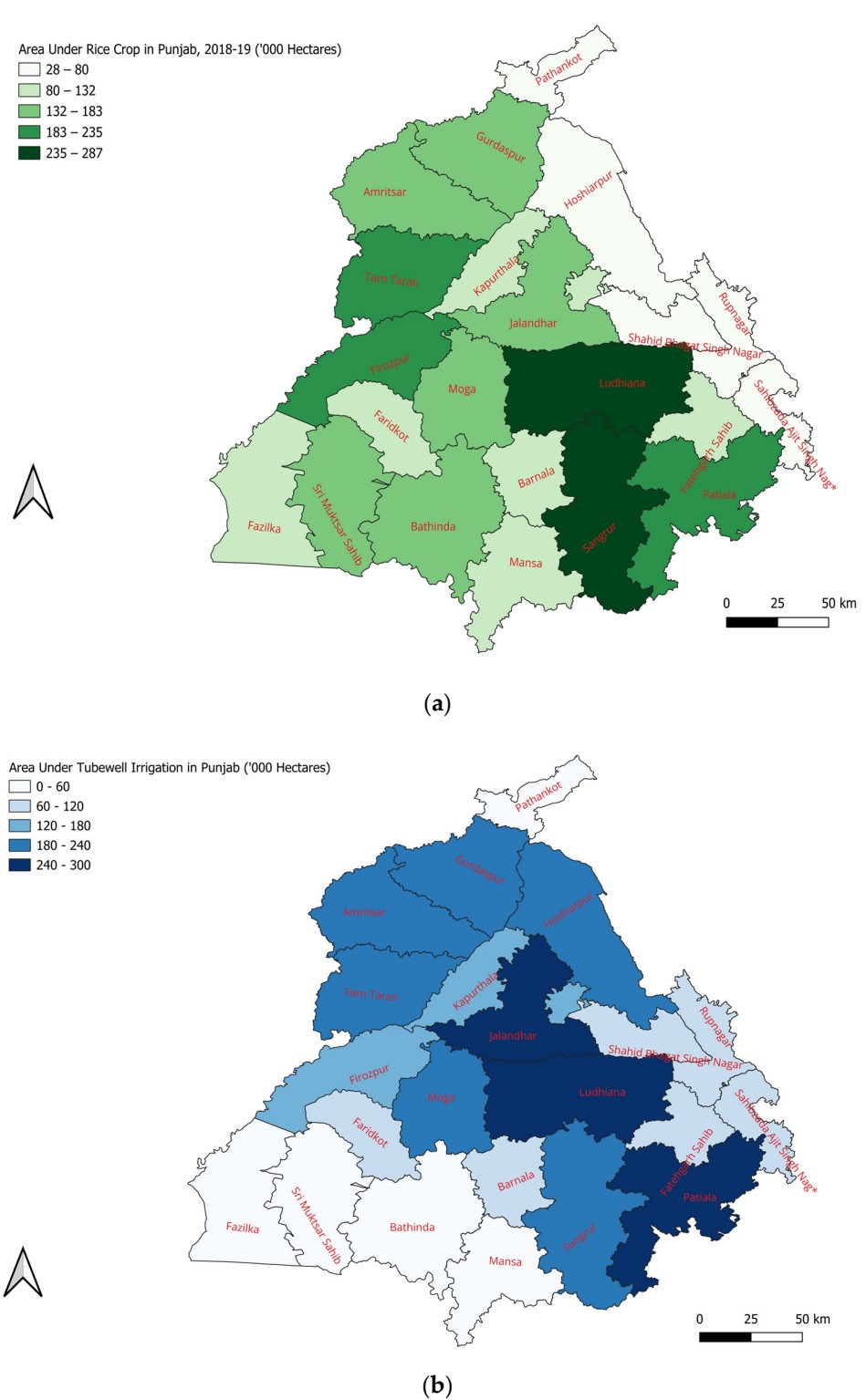

**Figure 5.** (**a**) Area under rice crop in Punjab, 2018–2019 ('0000 hectares); (**b**) area under tube well irrigation, 2018–2019 ('0000 hectares); source: authors' own mapping using Statistical Abstract of Punjab, 2020 [96].

### 3.3. The Punjab Water Resources (Management and Regulation) Act 2020

The state enacted the Punjab Water Resources (Management and Regulation) Act 2020 in February 2020, which established a much-needed Punjab Water Regulation and Development Authority (PWRDA), which makes the registration of new tube wells mandatory and checks on existing tube wells' usage necessary. It also establishes a fund in the name of the authority itself to meet various objectives of the Act. One of these objectives is the setting up of an Advisory Board consisting of multidisciplinary experts meeting every six months to advise the authority on water-related issues. A block-wise integrated state-water plan reviewed every two to three years is to be prepared by the authority. The authority may issue directions related to restrictions on the use of water, water conservation, and groundwater recharge for water users and advise on policies regarding the optimal utilization of irrigation potential, use of the latest technology, promotion of water-conservation awareness, increasing water-use efficiency in agriculture and other sectors, and preventing the contamination of water. It specifies the imposition of the tariff separately for the commercial/industrial sector and household/agriculture sector but will be notified in future orders [112].

While the Punjab Preservation of Subsoil Water Act 2009 primarily concentrated on addressing the overexploitation of groundwater related to paddy cultivation, the 2020 Act takes a broader and more holistic approach to water-resource management. The 2020 Act establishes the PWRDA, which serves as a comprehensive regulatory body for managing and regulating all aspects of water resources in Punjab. In contrast, the 2009 Act did not create a dedicated regulatory authority. The 2020 Act sets up an Advisory Board of experts and emphasizes the importance of block-wise integrated state-water planning, providing a structured and multidisciplinary approach to water management. The 2009 Act did not have these specific mechanisms. The 2020 Act introduced the mandatory registration of tube wells, ensuring better control and the regulation of groundwater usage, while the 2009 Act did not have such a provision. The 2020 Act goes beyond regulatory measures and outlines a comprehensive policy framework for water management, conservation, and technology adoption, covering various sectors, whereas the 2009 Act primarily focused on regulating paddy cultivation to conserve groundwater.

The latest Act establishes mechanisms for coordinated water-resource planning, development, and management to ensure sustainable use. However, it lacks ways to encourage and implement the participation and engagement of various stakeholders, including farmers, local communities, and civil society organizations, which are crucial for successful water management. Not much has been mentioned in regard to the availability of accurate and up-to-date data on water resources, including groundwater levels, water quality, and usage patterns. The availability of reliable and comprehensive data can be a limitation, as it requires robust monitoring systems and regular data collection. Another shortfall of the Act is the lack of specific incentives for farmers to actively participate in groundwater management and conservation efforts. While the Act may provide regulations and guidelines for groundwater extraction, it does not offer sufficient incentives or support mechanisms to encourage farmers to adopt sustainable practices or technologies to help manage and conserve groundwater resources. Additionally, it underestimates the importance of promoting awareness campaigns and educational programs to raise farmers' understanding of the importance of groundwater conservation and the benefits of adopting sustainable practices. These initiatives foster a culture of responsible water use among farmers and empower them to contribute to preserving groundwater resources.

### 3.4. Issues to Be Addressed While Using C&C Approach as a Policy Instrument

Studies have shown the CAC approach to be specific and, in such a way, less flexible and overly restricting on the users of the resource in question. Still, its popularity cannot be neglected in managing environmental resource degradation or, in this case, groundwater exploitation. Nevertheless, first, some issues in its applicability should be discussed.

While establishing a relationship between the policymakers or the regulators and the farmers, the interests of the public and the nation require the foremost priority. It implies avoiding the over-regulation of farmers to maintain ample livelihood for the farmers and food security for the state and nation overall. It will involve setting appropriate standards after key discussions with farmers and local communities at the village or block level. To facilitate any long-term behavior change in farmers, enforcement and timely inspections by the local authorities at the village and block level will be useful. These local authorities should be directly accountable to the policymakers at the state level. A major step towards developing accountability is defining user rights for groundwater usage, which may be divided by the number of hours of tube well operations on a single aquifer in the case of multiple users. Stable power transmission to farms may also benefit this measure as the first author's ground research found that in the Jalandhar district of Punjab, unstable power transmission leads to the overextraction of groundwater. These user rights should also accompany the responsibility of farmers to further monitor for any unlawful extraction and incentives for any such reporting or issues faced by the local or state authorities. Empirical studies have declared the CAC approach as an expensive and inefficient instrument that, on its own, generates less revenue than required to sustain it in the long run [44,113–117]. In such cases, encouraging farmers to self-monitor becomes extremely necessary to avoid high compliance costs [118,119].

In Punjab, where about 68 percent of operational land holdings are marginal, small, and semimedium, it is imperative to distinguish the strengths and weaknesses of such farmers from medium and large farmers [120]. The former set of farmers face different problems, and the motivation they would require to conserve groundwater would also be different. Since they form such a major portion of irrigators in the state, the total environmental impact of their actions on groundwater would be much higher and, on the other hand, would have lower compliance with the authoritative regime. Such farmers may lack resources, a sense of environmental awareness, and access to education and training. It was found during the first author's research in Jalandhar that the farmers did not know about the 'Punjab Preservation of Subsoil Water Act' and its objectives set by the state to conserve groundwater. These farmers fight for their basic survival and are unaware of reliable advice sources, and if they should respond with innovation regarding groundwater conservation, the right regulations with mutual agreements are necessary.

The laws implemented in Punjab concerning water management have been deemed insufficient and fall short of addressing the comprehensive measures necessary for effective groundwater-resource management. Despite various regulations and statutes, the current legal framework does not adequately tackle the complex challenges of water management in the region. There could be negative consequences of a fragmented approach to decision making, where different agencies are responsible for different aspects of water management, resulting in conflicting priorities and inefficient decision making [121,122]. Second, separating groundwater rights from land rights and adopting a comprehensive, integrated approach to resource management is needed. It is recommended that the water regulatory authority be established as a statutory body to oversee the management of water, ensuring substantial investment in irrigation infrastructure and the development of operational plans. However, until a comprehensive volumetric system is in place, alternative methods of pricing and regulating water usage may be considered. A multipronged approach is needed that includes the development of alternative water sources with emphasis on the need for greater awareness and participation among stakeholders, including farmers, policymakers, and the public, to address this pressing issue. Building institutional and monitoring capacity is also pivotal in effectively managing groundwater resources. Recognizing the significance of groundwater as a vital source of water supply, it becomes imperative to establish robust institutions and enhance monitoring mechanisms to ensure its sustainable use and protection. [68,123,124].

*3.5. An Approach to Regulating Groundwater Using CAC & I*

Marginal, small, and medium farmers can benefit financially from groundwater-resource conservation only by integrating sustainable groundwater practices, which can be imparted through long-term education and training. Periodic training can remove attitudinal obstacles related to ignorance and a lack of capability to manage groundwater, including underestimating the impact of their daily activities on the aquifers. This will depend on how information is dispensed to the farmers and who dispenses it. This will involve, firstly, describing to farmers how effective environmental practices in groundwater management will translate to economic gains for them. Secondly, farmer–state partnerships involving not only the in-person coordinated transmission of information but also ensuring that the information is well and effectively received while eliminating duplication. Third-party leverage involving banks, insurance companies, and input companies can be reliable sources of information for the farmers and encourage them to adopt aquifer-management practices. Such stakeholders can also be involved in periodic audits and environmental control checks and could impose certain clauses on farmers for the services provided to them. Education and training cannot be seen as a stand-alone method to regulate groundwater usage. It is a single component of a much larger integrated strategy, including controls, voluntary action, advice, and support for water-management technologies with the use of incentives.

While the traditional CAC approach uses direct regulation involving frequent ground-level audits by the concerned policymakers, given limited financial resources, a shift towards 'second best' techniques involving self-assessment and voluntary compliance is necessary. Hands-on training involving oral and written/pictographic explanations for farmers may enable them to identify and control groundwater exploitation and their responsibility for effective management. A periodic report outlining the challenges faced and measures, if any, taken, and the requirement of an external inspection, if necessary, may be submitted first at the village level and may be taken at the block/district level later. It will be, however, difficult to single out noncompliers from serious farmers who want to practice sustainable development. Farmers may be incentivized to take such actions in turn for protection from punitive action in the future if any corrective action is taken or suggested by the farmer. It should be clarified to the farmers that the only options are collaboration with the state with underlying incentives or fines and penalties. Large farmers are the ones who are most likely to adopt behavioral changes by having access to first-hand information. In such ways, they can influence the environmental behavior of small and medium farmers by forming partnerships.

There is a need for economical solutions to remove the limitations of small and medium farmers and, at the same time, be understandable, practical, achievable, and flexible. The state's role becomes essential in removing barriers to information, providing external subsidies if needed, internalizing externalities, publicly recognizing the best environmental practices, and providing technical assistance as different forms of incentives. A three-tier system of the CAC & I approach may also be initiated where the districts or blocks classified as overexploited in groundwater reserves may be required to have the highest control in the hands of the state in the form of licenses and permissions for every decision that may have an impact on aquifer depletion/recharge. Blocks and districts under semicritical and safe groundwater reserves can opt for self-regulation and management with the submission of periodic reports to the state, and those under critical reserves may fall anywhere between the two categories mentioned [38] where the state may rely on incentivization for complaints made against violations. The same is depicted in Figure 6.

The goal of any policy like CAC & I in the short term may imply bringing as many farmers as possible in compliance with groundwater conservation. However, the longer-term target is persuading farmers that reducing groundwater exploitation can lead to cost savings and a competitive advantage in the long run.

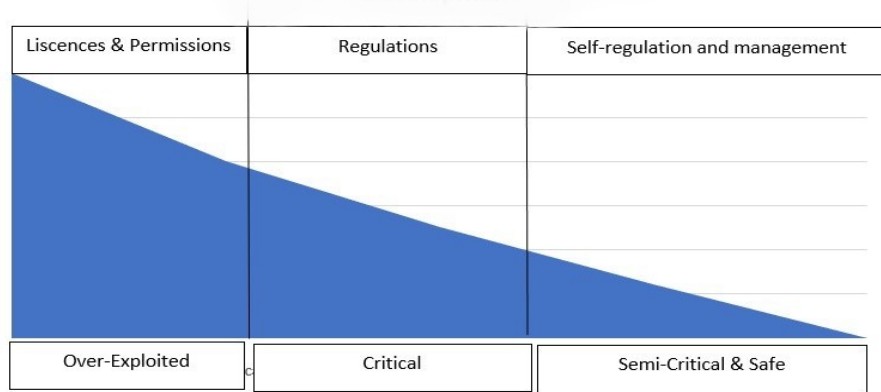

**Figure 6.** Author's representation of levels of control required based on the category of groundwater exploitation.

## 4. Discussion

The inadequacy of traditional common laws governing groundwater, which grant landowners absolute rights without requiring resource protection or replenishment, leads to the excessive personal use of groundwater and neglect of community needs despite the lack of effective legislation addressing the issue, as observed in India and the state of Punjab where groundwater rights are still governed by the Indian Easement Act of 1882, granting unlimited extraction rights tied to land ownership [3–8].

Additionally, the depletion of groundwater resources is attributed to advancements in geological understanding, drilling techniques, and pump technology, along with factors like population growth, urbanization, rising water-consumption rates, and climate change impacts. The Green Revolution in India further intensified groundwater extraction for irrigation, leading to adverse effects such as waterlogging, salinization, pollution, and declining water tables [16–21].

The global assessment of groundwater depletion indicates a significant increase in total global depletion from 1960 to 2000. Around 80% of the global population in 2000 resided in regions facing water security threats. Specific aquifers, including the Ganges, Indus Basin, Californian Central Valley, North China, and Tarim Basin, exhibit declining groundwater rates. Adverse consequences of depletion include reduced tube well yields, higher pumping costs, land subsidence, decreased base flow to surface water bodies, and saltwater intrusion. Unsustainable groundwater pumping affects food and water security, socioeconomic inequality, and cropping intensity [18,20,22–25,27–31].

Despite high rice yields and substantial production, Punjab in India faces food stress due to increasing water scarcity. Punjab's severe groundwater situation is evident in the 2022 CGWB report (164% groundwater development) characterized by 'Physical' scarcity, where demand surpasses supply [15,89–92]. Government policies promoting higher Minimum Support Prices (MSP) and power subsidies have led to a monoculture dominated by wheat and rice crops. The area under wheat and rice has steadily increased, while the diversification index has declined, indicating heavy reliance on a limited number of crops. During 2005–15, there was a decrease in the cultivation of alternative crops, such as pulses, oilseeds, fruits, and vegetables, contributing to a lower crop-diversification index. This decline raises concerns about the sustainability and resilience of agriculture in Punjab [30,68,70,74,75]. The lack of crop diversification leads to an over-reliance on water resources and hampers the overall agricultural productivity and sustainability in the region [79–81].

To address this issue, the Punjab Preservation of Subsoil Water Act 2009 was introduced as a legislative measure to manage and conserve groundwater resources. The Act primarily focuses on crop scheduling to reduce evapotranspiration and does not address demand management or other water-extraction technologies. While the Act focuses on crop scheduling to reduce evapotranspiration, it has not effectively addressed other extraction is-

sues. Studies indicate reduced depletion but also a rise in the tube well density, population, and higher depletion levels from 2009 to 2019. The proportion of areas with a water table depth over 10 m increased from 30% in 2000 to over 75% in 2019. Overall groundwater development decreased, but a district-wise analysis reveals mixed results, casting doubt on the Act's effectiveness. Overexploited blocks have shown little improvement.

Technologies like laser land leveling and conservation tillage methods can help reduce aquifer depletion and stubble burning in paddy cultivation. The government has allocated funds for promoting crop diversification and adopting water-saving methods. In addition, a new agriculture policy is planned to conserve natural resources and support tree planting. Still, groundwater depletion is an issue affecting farm productivity, as this paper has analyzed. Externalities of groundwater exploitation need to be addressed, including mass exploitation and increasing costs for small farmers.

Punjab's high agricultural productivity necessitates specific policies for small and marginal farmers. The Punjab Water Resources (Management and Regulation) Act 2020 complements the 2009 Act and provides a more comprehensive framework for water-resource management. However, it lacks provisions for stakeholder engagement, including farmers and local communities. Accurate and up-to-date data on water resources are also not adequately addressed. The Act does not incentivize farmers to participate in groundwater management and conservation actively. It also overlooks the importance of awareness campaigns and educational programs to promote responsible water use among farmers. Punjab's farmers, facing unique challenges, lack resources for groundwater conservation and may not fully grasp conservation regulations. Tailored approaches and mutual agreements are needed to encourage innovative groundwater-saving practices. Inadequate water-management laws, fragmented decision making, and the absence of integrated approaches are significant hurdles. Addressing these issues requires the establishment of a water regulatory authority, infrastructure investment, heightened stakeholder awareness, and enhanced institutional capacity and monitoring for sustainable groundwater use.

The CAC approach is commonly used to manage environmental resource degradation like groundwater exploitation. However, its effectiveness is context specific, while market mechanisms, economic incentives, and voluntary compliance through self-regulation are advocated as smarter approaches. Bottom-up groundwater governance, particularly in smallholder farmer contexts, has shown promise, utilizing market-based approaches with CAC measures or hybrid systems. Successful examples include conservation plans in Kansas, groundwater subdistricts in Colorado's San Luis Valley, and the Haryana model in India, offering incentives and restrictions on paddy cultivation in areas with deep groundwater depletion [37–39,50,53,57,59,62].

The limited understanding of the transferability of successful groundwater-governance schemes to stressed aquifer systems poses a challenge for developing conservation strategies, as applying similar rules widely may reduce institutional diversity, misalign with social and environmental contexts, and perpetuate recurring resource-management issues [50,63–66]. To address these issues, it is important to prioritize societal interests while avoiding the excessive regulation of farmers. Setting appropriate standards through discussions with farmers and local communities, along with enforcement and inspections by accountable local authorities, can promote behavioral change. Defining user rights for groundwater usage and encouraging self-monitoring by farmers can enhance accountability and reduce compliance costs.

Marginal, small, and medium farmers can benefit financially from groundwater-resource conservation through long-term education and training. Periodic training can address attitudinal obstacles and enhance farmers' capability to manage groundwater effectively. Farmer–state partnerships, involving the coordinated transmission of information and the involvement of third-party stakeholders, can support aquifer-management practices. Education and training should be integrated into a larger strategy that includes controls, voluntary actions, and incentives for water-management technologies. Shifting towards self-assessment and voluntary compliance, rather than direct regulation, can be

more feasible given limited financial resources. Hands-on training and periodic reporting can help farmers identify and control groundwater exploitation. Incentives can be provided to encourage sustainable practices and protect farmers from punitive action. Large farmers can play a role in influencing the environmental behavior of small and medium farmers through partnerships and information sharing. Economical solutions are needed to address the limitations of small and medium farmers in groundwater management. The state plays a crucial role in providing information, subsidies, internalizing externalities, recognizing best practices, and offering technical assistance as incentives. A three-tier CAC & I approach can be implemented, where districts or blocks with overexploited groundwater reserves have strict state control, semicritical and safe reserves allow for self-regulation with reporting to the state, and critical reserves fall somewhere in between. Incentives can be used to address violations and complaints.

## 5. Conclusions

The analysis of the Punjab Preservation of Subsoil Water Act 2009 and the Punjab Water Resources (Management and Regulation) Act 2020 shed light on the shortcomings in groundwater management in Punjab. The findings indicate that the 2009 Act primarily focuses on paddy-crop scheduling and fails to encourage farmers to utilize groundwater efficiently. Consequently, the heavy exploitation of groundwater resources, exacerbated by the water–energy nexus and inefficient state policies, continues to be a pressing concern.

The adoption of the Punjab Water Resources (Management and Regulation) Act 2020 represents a notable improvement over its predecessor. The Act introduces some essential measures to address groundwater-management issues, such as the licensing and regulation of groundwater extraction and the establishment of water-user associations. However, it falls short in terms of comprehensive groundwater-management planning and incentives that would encourage farmers to adopt sustainable water-usage practices. The Punjab Water Resources (Management and Regulation) Act 2020 should be complemented by more specific and comprehensive regulations. These should include clear guidelines for groundwater licensing, extraction limits, and monitoring mechanisms. The Act should also include provisions for effective stakeholder engagement, encouraging local farmers and communities to actively participate in groundwater management. Regular and updated data on water resources are essential. In addition to regulatory frameworks, incentives play a vital role in encouraging sustainable groundwater practices. Policymakers should introduce a range of financial incentives, subsidies, and rewards aimed at motivating farmers to adopt responsible water use. These incentives can be tied to specific actions, such as crop diversification, the adoption of water-efficient technologies, and conservation efforts.

Effective groundwater management also involves educating and raising awareness among farmers about the importance of sustainable water use. Implementing educational programs, workshops, and awareness campaigns will help farmers understand regulations, objectives for conservation, and best practices. This will be particularly beneficial for small and marginal landholders who may lack resources for groundwater conservation. Establishing water-user associations and fostering a sense of community ownership is crucial. Encouraging local communities or villages to actively participate in groundwater management and decision-making processes will lead to better stewardship and localized solutions. Ongoing monitoring and adaptation, continuous assessments of the impact of the multifaceted approach, and necessary adjustments based on data and feedback from the farming community are essential to ensure the long-term sustainability of groundwater resources.

This paper emphasizes that a CAC approach, on its own, can be costly, inflexible, and inefficient. However, when combined with incentives for farmers to practice self-regulation, it can pave the way for more-effective groundwater management. By providing incentives and support for adopting sustainable practices, such as promoting crop diversification, implementing water-efficient technologies, and rewarding conservation efforts, farmers can be encouraged to play an active role in groundwater management.

To achieve comprehensive and sustainable groundwater management in Punjab, it is essential to move beyond CAC measures and prioritize the integration of incentives that align with the practical needs and circumstances of the farming community. A combination of regulatory measures, participatory approaches, and financial incentives can foster a sense of ownership, responsibility, and stewardship among farmers. This, in turn, can lead to efficient groundwater utilization and long-term sustainability.

**Author Contributions:** Conceptualization, S.B. and S.P.S.; methodology, S.B.; investigation, S.B.; resources, S.B. and S.P.S.; data curation, S.B.; writing—original draft preparation, S.B.; writing—review and editing, S.P.S.; visualization, S.P.S.; supervision, S.P.S. All authors have read and agreed to the published version of the manuscript.

**Funding:** This research received no external funding.

**Institutional Review Board Statement:** Not applicable.

**Informed Consent Statement:** Not applicable.

**Data Availability Statement:** Publicly available datasets were analyzed in this study. This data can be found here: [1. http://cgwb.gov.in 2. https://aps.dac.gov.in 3. https://punjab.gov.in 4. https://www.indiabudget.gov.in].

**Conflicts of Interest:** The authors declare no conflict of interest.

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
