# Peer review of "Can an Incentivized Command-and-Control Approach Improve Groundwater Management? An Analysis of Indian Punjab"

_sustainability, doi:10.3390/su152215777_

Round 1
Reviewer 1 Report
Comments and Suggestions for Authors
I appreciate the valuable insights presented in the research paper titled "Implications of Groundwater Depletion and Legislation: A Case Study of Punjab, India." The study sheds light on a critical issue and offers potential strategies for sustainable groundwater management. While the paper is well-structured and informative, I have a few queries and suggestions to enhance its clarity and impact:
1. The abstract provides a clear overview of the research objectives and methodology. It effectively highlights the issue of groundwater depletion in Punjab, India, and the significance of the Punjab Preservation of Subsoil Water Act 2009. The inclusion of qualitative research methods, such as GIS mapping, adds credibility to the study. However, the abstract could benefit from specifying some of the key findings or insights derived from the analysis. This would provide a glimpse into the paper's contribution and make it more enticing for readers. Additionally, a concise mention of the limitations encountered during the research process could enhance the transparency of the study.
2. The introduction could be further strengthened by providing a clear research gap or question that the paper aims to address. While the broad topic is introduced comprehensively, a specific focus for the study could be explicitly stated. This would help readers understand the unique contribution of the paper within the larger context of groundwater management literature.
3. The introduction introduces the concept of command-and-control (CAC) regulations and discusses their limitations, but it could elaborate further on how these limitations apply to groundwater management specifically. This could set the stage for the discussion on incentivized approaches later in the paper.
4. The use of primary survey data conducted in four districts of Punjab adds credibility to the study's findings. However, it would be beneficial to elaborate further on the sampling strategy employed for the primary survey. Details regarding how participants were selected, their representativeness, and any potential biases should be included to enhance the transparency of the study.
5. The Discussion section seems to be densely packed with citations. While the comprehensive referencing strengthens the foundation of the discussion, some key points could be highlighted and discussed in greater detail, showcasing the paper's unique contribution. The discussion could also benefit from a more focused connection to the paper's research objectives and findings, emphasizing how the identified challenges tie into the research question.
6. To offer a more comprehensive understanding, could you expand on the key provisions of the 2020 Act and provide insights into how it compares or contrasts with the Punjab Preservation of Subsoil Water Act 2009?
7. While you've outlined the challenges in groundwater management, incorporating real-world examples or case studies illustrating these challenges would strengthen your arguments.
8. While your conclusion effectively summarizes key findings, could you expand on the proposed multifaceted approach for comprehensive groundwater management? This would provide a clear pathway for future actions.
9. In discussing global groundwater depletion, could you provide more recent statistics beyond 2000 to ensure the paper's current Ness?
10. To improve the flow of the content, consider ensuring smoother transitions between different sections of the discussion.
11. The image in Figure 5 appears faded and lacks clarity. Please replace it with a clear and well-defined image.
12 In references 25 and 29, kindly provide the full list of authors for better citation accuracy.
13.. Could you please include the web address for Reference 15 to enhance the accessibility of the source?
Comments on the Quality of English Language
1. Some paragraphs in the discussion section are quite long and contain multiple ideas. Consider breaking them into shorter paragraphs to improve readability and coherence. For instance, the paragraph that starts with "The global assessment of groundwater depletion indicates a significant increase..." can be split into smaller paragraphs, each focusing on a specific aspect.
2.The paper effectively employs academic language; however, ensure that the vocabulary aligns with the intended audience. Complex terminology should be explained or contextualized for readers who might be less familiar with the subject matter.
Reviewer 2 Report
Comments and Suggestions for Authors
The article topic is really interesting; however, the paper has some problems that should be addressed to be considered for publication.
1) The quality of the figures should be improved. Fig 1 (It is not necessary the title “Area Under Major Crops in Punjab, 1980-2020” on the figure since it is written in the figure title). Fig 2, 3, 4 and 5 should be improved (we can’t see the names of the regions and the quality is too low). Fig 6 (The title “Levels of control based on the category of groundwater development” is not necessary since it is written in the figure title and the quality is low). Fig 2, 3, and 5 should be divided into (a) and (b) included in the Figure title. It will facilitate the analysis of the figure.
2) Table 2 is confusing. Please, place the year (2009/2017) in a column.
3) Why are you using 2017 data? Is there the newest data? If so, add a third year to improve the discussion. 2017 data is quite old.
4) Please, add a section with the requirements of Punjab Preservation of Subsoil Water Act 2009 for the readers to understand what is this legislation about.
5) Please, create a section very concise on how this Water Act should be improved. The authors should be more critical about it, considering the analysis of the paper.
Round 2
Reviewer 1 Report
Comments and Suggestions for Authors
1. In references 2, eabd2849 given, what it indicates?
2. Given web address for Reference 15 link shows "Page not found", can you put correct web address.
Reviewer 2 Report
Comments and Suggestions for Authors
The quality of the manuscript has improved.
